# Using a Motivational Paradigm to Develop an Exercise Program for Nurses with High Risk of Metabolic Syndrome

**DOI:** 10.3390/healthcare11010005

**Published:** 2022-12-20

**Authors:** Wen-Ping Lee, Pao-Yuan Wu, Li-Chin Chen, Whei-Mei Shih

**Affiliations:** 1Department of Nursing, New Taipei Municipal Tucheng Hospital (Built and Operated by Chang Gung Medical Foundation), New Taipei City 236, Taiwan; 2Department of Nursing, University of Kang Ning, Taipei City 114, Taiwan; 3Department of Nursing, Chang Gung University of Science and Technology, Taoyuan 333, Taiwan; 4Graduate Institute of Gerontology and Health Care Management, Chang Gung University of Science and Technology, Taoyuan 333, Taiwan

**Keywords:** transtheoretical model, exercise program, metabolic syndrome, nurse

## Abstract

Nurses are frontline care providers whose health is vital to providing good quality of care to patients. The purpose of this study was to develop an exercise program for high-risk metabolic syndrome nurses based on the transtheoretical model. The transtheoretical model was used in this study due to its popular use in exercise behavior change and it can clearly identify the stage of exercise so as to plan an effective program to promote health. This was a quasi-experimental pilot study with a total of 40 participants who met the inclusion criteria. Exercise programs were developed for three groups distinguished by their commitment to exercising for health. Sixteen (40%) nurses moved one step forward, six (15%) nurses moved backward, and eighteen (45%) nurses maintained at the same stage over time (stable sedentary, 40%; stable active, 5%). Bowker’s test of symmetry, χ2 = 14.00 (*p* < 0.01), revealed that the population exercising increased significantly after the intervention. After the program, the perceived benefits from exercise in the decisional balance significantly increased to 1.53 (t = 2.223, *p* < 0.05), perceived exercise barriers significantly decreased to 3.10 (t = −3.075, *p* < 0.05), and self-efficacy significantly increased to 2.90 (t = 3.251, *p* < 0.01), respectively. Applying the transtheoretical model to health behavior enables significant change. The benefits of applying the transtheoretical model for promoting exercise include increasing perceived exercise benefits and self-efficacy, decreasing perceived exercise barriers, and increasing physical activity levels.

## 1. Introduction

Nurses are not only crucial to the delivery of clinical care; they also act as ‘healthy’ role models for patients regarding normal weight and healthy behavior. Irregular working hours due to shift work cause disturbance of the circadian rhythm, leading to sleep deficits, physical discomfort, and changes in diet and mealtimes. Together with the heavy workload and stress, these working conditions disrupt the body’s metabolism resulting in carbohydrate tolerance and increased blood cholesterol, triglycerides, and low-density lipoprotein cholesterol [1,2].

Such metabolic changes are particularly evident among shift workers when compared with day workers [1,3,4]. Moreover, studies have found that metabolic syndrome (MetS) is associated with coronary heart disease and type 2 diabetes mellitus and has also been linked to a two-fold increase in cardiovascular motility [5,6]. In addition, regular exercise has proven to be beneficial to physical and mental health, as it helps minimize risk factors for chronic diseases, reduces morbidity and mortality, saves medical expenses, lowers social costs, and prevents waste [7,8,9,10].

Studies on the prevalence of MetS among nurses have reported incidences ranging from 5.7% to 38.7% among nurses whose average ages ranged from 37.9 to 44 years old internationally [2,11,12]. In Taiwan, the incidence of MetS ranged from 5.0% to 13.84% among nurses with an average age of approximately 35 years old [13,14,15]. Additionally, younger groups of nurses in Taiwan had higher incidence of MetS, suggesting that shift work, heavy workloads, and irregular life patterns are likely risk factors for MetS [14]. Furthermore, nurses in Taiwan are less likely to engage in regular exercise as compared with nurses elsewhere, ranging from 11.1 to 26.7%, whereas the exercise rate ranged from 30 to 42% in other studies [16,17,18,19]. Compared with other countries, nurses in Taiwan are less fond of exercise which leads to harmful effects to health. 

There are many theories of behavior change such as the health belief model, cognitive behavior change model, precede and proceed model, and transtheoretical model (TTM), among which the TTM is one of the most commonly used theoretical frameworks for health promotion in physical exercise [20,21,22]. It is a simple and effective model for classifying a target population and planning interventions. The model involves several aspects of behavior change, including cognition, behavior, as well as time [23]. TTM components include stages of change (indicating people’s motivation level), processes of change (strategies for modifying behavior), decisional balance (perception of pros and cons of a behavior), and self-efficacy (confidence about achieving a specific behavior). 

The first of these stages of change is an important TTM construct for identifying the individuals’ position in the process of change. The process of behavior change involves five different stages, namely (1) precontemplation: no intention to start exercising in the next six months; (2) contemplation: thinking about starting to exercise in the next six months; (3) preparation: currently exercise some, but not regularly; (4) action: currently exercise regularly, but have only begun doing so within the last six months; and (5) maintenance: currently exercise regularly and have done so for longer than six months. Regular exercise refers to three times or more per week for 20 min or longer. The process of behavior change is not always a linear progression through the stages; individuals may also move through the stages in a spiral pattern. For example, before reaching the maintenance stage, individuals may try different types of behavior change and can regress to an earlier stage (e.g., return from the action stage to the contemplation stage), forming a dynamic forward or backward spiral pattern of movement [23,24,25].

The second key TTM construct is processes of change, which are postulated to explain whether or how people influence their experiences and environment to further modify their behavior. Processes of change are categorized as cognitive or behavioral. The cognitive processes consist of consciousness-raising (awareness about a behavior), dramatic relief or emotional arousal (affective aspect of a behavior), environmental re-evaluation (how the problem influences the environment), self-re-evaluation (emotional and cognitive appraisal of the impact of the behavior in terms of its value for the individual), and social liberation (awareness about social opportunity and alternatives). The behavioral processes are self-liberation (choosing and committing to change), managing reinforcement (self-reward for change), helping relationships (enlisting assistance or social support), counter-conditioning (substituting alternative behaviors for the problem behavior), and stimulus control (removing cues for unhealthy habits) [23].

The third TTM construct posits a decision-making balance in which pros outweigh cons for deciding whether or not to adopt a new behavior. When the pros are greater than the cons, the transformation moves from the precontemplation to the contemplation stage [23,24]. The fourth construct is self-efficacy, which is derived from Bandura’s theory [26].

Abundant research has proven TTM to be effective for changing exercise behavior; the approach has been applied successfully to different groups, including students, obese people, diabetics, and people with cancer [27,28,29,30]. However, to our knowledge, exercise programs for MetS for different stages were not provided. This study investigated the effectiveness of individualized exercise programs based on the transtheoretical model, with the goal of increasing physical activity and improving indicators of MetS among nurses. 

## 2. Materials and Methods

This is a pre-experimental single-group intervention pilot study using TTM in a medical center of northern part of Taiwan. This study adopted the definition of MetS described by the National Cholesterol Education Program. MetS is a clustering of at least three of the five following medical conditions: abdominal obesity, defined as waist size of ≥40 in (≥102 cm) for men and ≥35 in (≥88 cm) for women, plasma triglyceride (TG): ≥150 mg/dL, serum high-density lipoprotein (HDL) of <40 mg/dL for men and <50 mg/dL for women, blood pressure of ≥130/85, and fasting plasma glucose of ≥110 mg/dL [31].

Since the purpose of this study was to prevent participants from progressing to additional MetS risk factors, a total of 50 participants with one or more metabolic risk factors were identified from employee health examinations and were recruited for this study, assuming an effect size of 0.4, power of 0.8, and 20% attrition rate. Inclusion criteria specified that the participating nurses must be female, over 20 years old, have worked more than one year, and meet at least one of the criteria for MetS (out of 5 risk factors) according to the NCEP. Exclusion criteria included part-time nurses, male nurses, and pregnant nurses. 

Out of 46 participants, 4 participants in group 1, 1 participant in group 2, and 1 participant in group 3 were excluded from the study due to pregnancy, refusal to participate, and withdrawing from the study, leaving 40 participants in total. The study was completed with 13 participants in group 1, 20 participants in group 2, and 7 participants in group 3. Figure 1 shows the flow diagram of the study.

This study received ethical approval from the hospital (104-0998C) and all participants provided written informed consent. 

### 2.1. Data Collection Tool

(1) Stages of Exercise Behavior Change Scale.

We used the Chinese version of the Stages of Exercise Scale (SOES) developed by Kao et al. [32]. Test-retest reliability at two-week intervals were 0.81 and 0.82, respectively, indicating that the SOES had good reliability. The test-retest reliability for this study was 0.91 and 0.92, respectively. 

(2) The Perceived Exercise Benefits Scale, the Perceived Exercise Barriers Scale, and the Exercise Self-Efficacy Scale.

All three scales were modified by Kao et al. [32], with an 18-question Perceived Exercise Benefits Scale (PEBS-18), a 19-question Perceived Exercise Barriers Scale (PEBS-19), and a 19-question Exercise Self-Efficacy Scale (ESES-19). Higher scores on the three tests were associated with stronger belief that exercise is beneficial (PEBS-18), with perceiving barriers to exercise as more severe (PEBS-19), and with greater self-confidence about overcoming the obstacles and maintaining regular exercise (ESES-19), respectively. The Cronbach’s α values of the official questionnaire for these three scales were 0.96 (PEBS-18), 0.87 (PEBS-19), and 0.96 (ESES-19), respectively. The test-retest reliability for this study was 0.93, 0.84, and 0.92. 

### 2.2. Program Design

A meta-analysis of randomized controlled trial studies suggested that interventions shorter than 14 weeks could maximize the increase in physical activity behavior [33]. Therefore, we provided a stage-specific, TTM-based exercise behavior modification program for nurses at different stages of behavioral change for 12 weeks. The program was held in a hospital gymnastic center. Before the program was given, each participant filled out the SOES to allocate them to different stages. The program included teaching activities, stretching exercises, and group discussions. Participants were divided into three groups based on the distribution of exercise stages identified by the pilot study.

Group one, “Get Moving,” was for nurses in the precontemplation and contemplation stage. In this stage, the strategies were as follows: (1) Administer health exam (e.g., blood test, blood pressure, and body mass index) for the individuals to understand their own health status and to promote self-awareness in healthcare. (2) Design in-service exercise education programs that concentrate on knowledge about exercise and guiding principles for physical exercise in order to promote understanding of the physical health benefits of exercise, such as teaching how to relive muscle soreness after physical activity and preventing sports injuries, with the aim of increasing intention to exercise. (3) Organize educational lectures on nutrition to improve knowledge of what constitutes a healthy diet, including learning how to convert food calories, reference intake for different nutrients and food choices, and avoiding processed foods, as well as a high-sugar and high-fat diet. (4) Launch regular lectures on exercise and nutrition to promote knowledge about physical activity, balanced diet, calorie calculation, and healthy tips for eating out; to strengthen participants’ understanding of the benefits of exercise and balanced diet; and to promote knowledge about the consequences of lack of physical activity and unbalanced diets, so that participants are aware of the importance of exercise and to motivate them to adopt exercise behavior. (5) Encourage the participants to use exercise resources available at the workplace (such as gym equipment and training programs for staff only). (6) Discuss with nurses’ families to reach agreement on exercise engagement through better time management, e.g., compiling a housework to-do list and distributing the work based on family members’ ability to share household duties in order to help the nurses to create spare time to exercise. (7) Provide a family fitness program and gym with a playroom for nurses with young children. (8) Design a wide variety of sports programs with flexible timing for nurses performing shift work, such as morning jogging or indoor yoga. (9) Share experiences of exercise benefits with colleagues and offer individualized advice to encourage nurses to engage in exercise behavior. 

Group two, “Deciding to Plan,” was for nurses in the preparation stage. In this stage, the strategies were as follows: (1) Provide the nurses with a wide variety of sports resources to enhance their exercise self-efficacy, particularly low-to medium-intensity exercises that require low sports skills, that can be practiced alone, or that require fewer players, and that can become lifelong habits. (2) Provide time management courses to guide the nurses on optimizing their time in order to perform physical activity on a regular basis and finding the most suitable types of exercise and the best time for performing them. (3) Recommend assistive exercise devices, such as using a pedometer to record daily steps while setting a goal in terms of step numbers, choosing activities on which participants scored a confidence rating higher than 7, and helping them to motivate to walk more and perform more physical activity. (4) Encourage commitment to behavior change with positive reinforcement to strengthen the confidence in taking action, such as making a public declaration or making a birthday wish to encourage exercise engagement to foster determination and increase the likelihood of behavior change. (5) Support peers and family to provide reminders and to be exercise companions, as well as role modeling which can effectively trigger self-re-evaluation and moving toward a model of healthier life. 

Group three, “Keep Going,” was for nurses in the action and maintenance stages. In this stage, the strategies were as follows: (1) Promote stair climbing with rewards of food vouchers and offer special discounts for healthy set meals designed by dietitians in order to support regular exercise and a healthy diet at the workplace. (2) Create an exercise-friendly environment and social atmosphere, encourage support from family, colleagues, and management, and provide exercise equipment and resources at the workplace. (3) Provide a user-friendly environment, such as setting up a well-equipped 24 h office gym providing fitness professionals and offering exercise guides and physical training tips. (4) Hold regular group discussions with colleagues who habitually engage in physical activity to share their experience and give advice. (5) Use cues to remind participants about practices that support habit formation, such as marking the workout time on the calendar or setting mobile phone reminders for a fitness program. Send exercise-related information to nurses via work emails to help shift workers acquire new knowledge regarding the health benefits of sports. Help overcome the time and space limitations related to exercise, promote the distribution of health information, and encourage exercise behavior change. (6) Organize diversified groups and activities to promote sustainable exercise habits and enhance self-efficacy via group support and constraints, such as setting rules for rewards and punishments to foster positive behavior change and avoid relapse. (7) Organize workplace or community running and walking events or other health activities and evaluate health status (e.g., measurement of body fat, waist-hip ratio, and body mass index). (8) For nurses with low self-efficacy regarding exercise, sharing successful experiences with other nurses in the maintenance stage may enhance their confidence and sense of achievement, and further improve self-efficacy and promote regular exercise behavior. Encouraging self-care, participation in peer exercise programs and action-packed family activities enhance the overall social support for exercising. The exercise program design is shown in Table 1, Table 2 and Table 3.

### 2.3. Statistical Analysis

For distribution of relapsing and progressing between stages on pre-test and post-test, the study used a Bowker’s test of symmetry method from SPSS version 20. For comparison of decisional balance and self-efficacy between pre- and post-test, a paired *t*-test was used.

## 3. Results

### 3.1. Demographic Variables of Participants

A total of 40 participants completed the 12-week program, leaving 13 in group one, 20 in group two, and 7 in group three. Nurses’ ages ranged from 27 to 52 years, averaging 39.40 ± 8.51. Sixty percent (*n* = 24) of nurses were college graduates and all were working shifts (100%). As far as risk factors for MetS, 11 (27.5%) of the participants had one risk factor, 20 (50.0%) had two risk factors, and 9 had risk factors of ≥3 (22.5%). 

### 3.2. Analysis of Nurses Relapsing and Progressing between Stages

The results showed the exercise behavior change of each stage indicating more nurses moved forward to the next stage. To test the distribution of relapsing and progressing between stages on pre- and post-testing, we used Bowker’s test of symmetry method. Results revealed a significant difference between pre- and post-test as χ2 = 14.00, df = 5, *p* < 0.016 (Table 4). Nurses in the precontemplation stage decreased from 6 (15%) to 3 (7.5%); those in the contemplation stage decreased from 7 (17.5%) to 3 (7.5%); nurses in the preparation stage increased from 20 (50%) to 21 (52.5%); nurses in the action stage increased from 4 (10%) to 6 (15%); and those in the maintenance stage increased from 3 (7.5%) to 7 (17.5%) (Table 5).

Bock et al. [34] and Marcus et al. [35] categorized participants who moved one or more steps forward as adopters, while participants who moved one or more steps backward were categorized as relapsers. When participants remained at the same stage, whether precontemplation, contemplation, or preparation, they were considered stable sedentary. When they remained at the action and maintenance stages, they were classified as stable active. Comparing the pre-test and post-test results, overall, 16 (40%) nurses moved one step forward, 6 (15%) nurses moved backward, and 18 (45%) did not change over time (stable sedentary, 40%; stable active, 5%) instead maintaining at the same stage. When combining nurses in the action and maintenance stages to evaluate their participation in regular exercise, we observed a rate of 17.5% on pre-test and 32.5% on post-test, for a total increase of 15% in regular exercise. 

After the program was implemented, the perceived benefits of exercise during decisional balance increased significantly to 1.525 (t = 2.223, *p* < 0.05), perceived barriers to exercise significantly decreased to 3.075 (t = −2.423, *p* < 0.05), and self-efficacy increased significantly to 2.900 (t = 3.251, *p* < 0.01), as shown in Table 6.

## 4. Discussion

The results of this study using TTM to design exercise programs for nurses at high risk of MetS show that 16 participants (40%) moved one stage forward, for an increase of 15% in the exercising population. Though the percentage of participants exercising was lower in this study than in some other studies, the program still had a positive effect [3,32]. Only 16.7% of participants remained in the precontemplation stage. Seventy percent of participants in the precontemplation stage moved to the preparation stage. In contrast, 33.3% of participants remained in the maintenance stage. One participant moved back from the maintenance stage to the action stage and the other one moved back to the preparation stage. These shifts indicate changes in exercise stage towards becoming a regular exerciser. Our findings are consistent with those of other studies that observed that interventions based on TTM combined with a program of physical activities are effective in moving people toward a more active lifestyle, consistently promoting the perceived advantages of changing the behavior in the decisional balance and to taking action to exercise [5,23]. However, another study found no significant difference in the effect on individuals’ physical activity six months after the educational intervention. It is worth noting that the latter study did not apply any model for their intervention [36]. 

Regular exercise can prevent and control chronic diseases and studies have proven that the transtheoretical model can significantly help to improve regular exercise behavior. For example, in a study conducted in Taiwan, 193 subjects participated in a 24-week exercise program with the result that perceived benefits of exercise and exercise self-efficacy scores were significantly higher and perceived barriers to exercise decreased in the experimental group. The population of regular exercisers increased by 20% after the program was implemented [32]. Similarly, an intervention study by Dallow and Anderson compared 58 women classified as sedentary and obese to a control group. The intervention group participated in a 24-week physical activity program. The control group received only a traditionally structured exercise program. The treatment group showed significant improvement on the eight processes of change, as well as self-efficacy, physical activity, and cardiorespiratory fitness [37]. In addition, a study of chronic kidney disease patients by Chen et al. [38] provided a one-month group exercise program and telephone counseling during the second and third months. Results revealed significant differences in cholesterol level (198.03 ± 43.52 mg/dL to 160.97 ± 37.39 mg/dL) and an increase in the number of regular exercisers from 57.8% to 75.6% of the total participants. Other studies assessing a TTM-based educational intervention have likewise observed an increase in physical activity level in the intervention group as well as progress on the stages of change [5,39].

With our TTM-based program, we observed an increase in physical activity and change of stages consistent with the results obtained by Mostafavi et al. [5] and Moeini et al. [30], which also aimed at improving women’s physical activity and decreasing risk factors for MetS. 

In conclusion, TTM is a framework for assessing and addressing readiness for behavioral changes. This model has been validated in numerous studies of exercise [5,9]. The study suggests evidence by using TTM to promote nurses’ behavioral changes in exercise. 

Results from this study cannot be generalized since this pilot study had a small sample size limited to female participants and lacked a control group and measurement of the processes of change. Future studies should include male participants, a control group, and a larger sample size [40]. Since most of the nurses perform shift work, it is difficult for them to meet the time to participate in the exercise program. It is recommended to provide flexible exercise time to promote attendance. 

## 5. Conclusions

In conclusion, most of the studies regarding promoting MetS used one exercise program such as treadmill exercise training, swimming, yoga, and aerobic exercise as intervention [41,42,43,44]. The main contribution of this study is to provide clear exercise strategies according to each stage of change so as for nurses to develop regular exercise behavior and promote health. This study suggests that using the TTM has potential value for developing exercise programs for nurses at high risk of MetS. By using stages of change in TTM for assessing physical activity, it is easy to identify which stage nurses were in so as to develop an individually tailored program. TTM can be applied in different contexts, including communities, schools, and workplaces for behavior change planning. The benefits of applying TTM include increasing perceived benefits from exercise and increased self-efficacy, decreased perceived exercise barriers, and increase in physical activity levels. It is worth applying the ideas in this study to other workplaces. 

## Figures and Tables

**Figure 1 healthcare-11-00005-f001:**
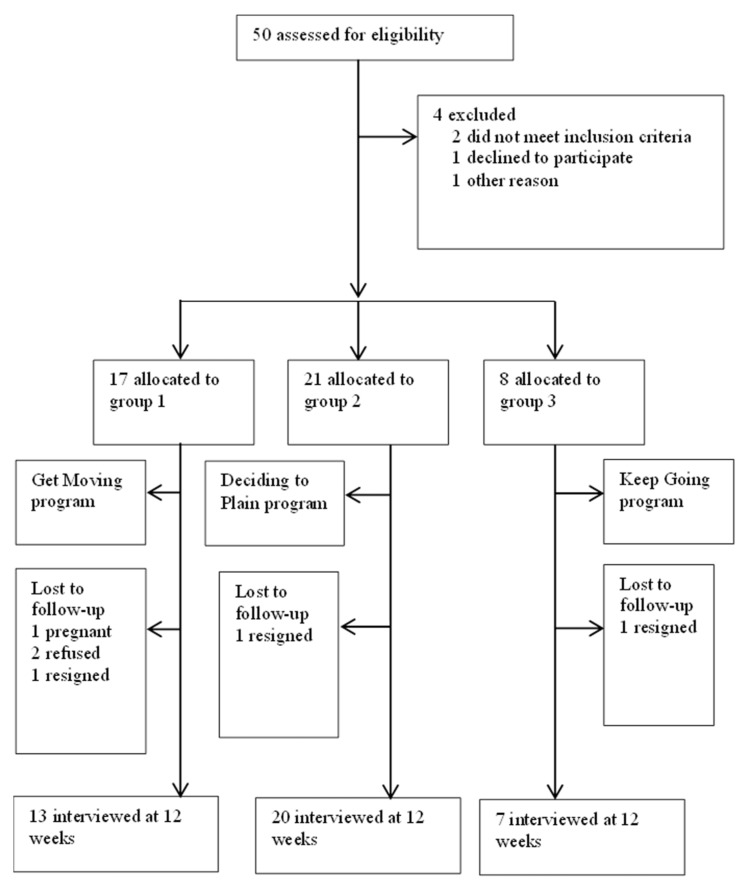
Flow diagram of the study.

**Table 1 healthcare-11-00005-t001:** Teaching strategies for group one—“Get Moving”.

Group One	Get Moving
Subjects	Nurses in precontemplation and contemplation stages
Goal	1. To arouse nurses’ will or action to exercise by providing knowledge about exercise and health. 2. To promote nurses transitioning from precontemplation and contemplation to the preparation and action stages.
Process of Change	Strategies
1. Consciousness Raising 2. Dramatic Relief 3. Environmental Reevaluation 4. Self-Reevaluation 5. Social Liberation 6. Self-Liberation	1. Administer health exam (e.g., blood test, blood pressure, and body mass index). 2. Design in-service exercise education programs. 3. Organize educational lectures on nutrition. 4. Launch regular lectures on exercise and nutrition. 5. Encourage the participants to use exercise resources available at the workplace. 6. Discuss with nurses’ families to reach agreement on exercise engagement. 7. Provide a family fitness program and gym with a playroom. 8. Design a wide variety of sports programs with flexible timing. 9. Share experiences of exercise benefits.

**Table 2 healthcare-11-00005-t002:** Teaching strategies for group two—“Deciding to Plan”.

Group Two	Deciding to Plan
Object	Nurses in the preparation stage
Goal	To consult exercise plan, to provide exercise methods and skills, and to develop individual exercise prescriptions so as to induce exercise behavior and decrease obstacles. To promote nurses from moving from the preparation stage to the action and maintenance stages.
Processes of Change	Strategies
Self-Re-evaluation Self-Liberation Helping Relationship	1. Provide the nurses with a wide variety of sports resources. 2. Provide time management courses. 3. Recommend assistive exercise devices. 4. Encourage commitment to behavior change. 5. Support peers and family.

**Table 3 healthcare-11-00005-t003:** Teaching strategies for group three—“Keep Going”.

Group Three	Keep Going
Object	Nurses in the action and maintenance stages
Goal	To maintain exercise motivation and will through exercise experience sharing and group discussion, so as to prevent reverting to previous stages and to build up a lifelong exercise habit.
Processes of Change	Strategies
1. Reinforcement Management 2. Helping Relationships 3. Counter Conditioning 4. Stimulus Control	1. Promote stair climbing. 2. Create an exercise-friendly environment and social atmosphere. 3. Provide a user-friendly environment. 4. Hold regular group discussions with colleagues. 5. Use cues to remind participants about practices. 6. Organize diversified groups and activities. 7. Organize workplace or community running and walking events or other health activities. 8. Share successful experiences with other nurses.

**Table 4 healthcare-11-00005-t004:** The distribution of nurses relapsing and progressing between stages on pre-test and post-test (N = 40).

	Post-Test Stages	PC	C	P	A	M
Pre-Test Stages (n)		n (%)	n (%)	n (%)	n (%)	n (%)
PC (6)		1 (16.7)	1 (16.7)	2 (33.3)	1 (16.7)	1 (16.7)
C (7)		1 (14.3)	1 (14.3)	3 (42.9)	1 (14.3)	1 (14.3)
P (20)		1 (5.0)	1 (5.0)	14 (70)	2 (10.0)	2 (10.0)
A (4)		0	0	1 (25.0)	1 (25.0)	2 (50.0)
M (3)		0	0	1 (33.3)	1 (33.3)	1 (33.3)
40 (100.0%)		3 (7.5)	3 (7.5)	21 (52.5)	6 (15)	7 (17.5)

This is a Bowker’s test of symmetry χ2 = 14.00. PC = precontemplation, C = contemplation, P = preparation, A = action, M = maintenance.

**Table 5 healthcare-11-00005-t005:** Change of TTM stage (N = 40).

	Stages	PC	C	P	A	M
Pre- and Post-Test						
Pre-test stages (n)		6	7	20	4	3
Post-test stages (n)		3	3	21	6	7
Changes of numbers		−3	−4	+1	+2	+4

PC = precontemplation, C = contemplation, P = preparation, A = action, M = maintenance.

**Table 6 healthcare-11-00005-t006:** Comparing decisional balance and self-efficacy between pre- and post-test (N = 40).

Variables	Pre-Test		Post-Test		Difference Average	t
	Mean	SD	Mean	SD		
Decisional Balance						
Perceived Exercise Benefits	79.85	8.652	81.38	6.86	1.525	2.223 *
Perceived Exercise Barriers	75.93	12.952	72.85	14.484	−3.075	−2.423 *
Self-efficacy	42.5	14.944	45.4	14.096	2.900	3.251 **

* Significant at the *p* < 0.05 level. ** Significant at the *p* < 0.01 level.

## Data Availability

The datasets used and/or analyzed during the current study are available from the corresponding author upon reasonable request.

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
