# Peer review of "Using a Motivational Paradigm to Develop an Exercise Program for Nurses with High Risk of Metabolic Syndrome"

_healthcare, 2022, doi:10.3390/healthcare11010005_

Round 1
Reviewer 1 Report
Thank you to the author for the present study "Using a Motivational Paradigm to Develop an Exercise Program for Nurses with High Risk of Metabolic Syndrome". The article is interesting and well written, however, there are some point to imporve.
First of all, the title of the article is not exactly in line with the study, please fix it.
In some points there is a lack of references, such as in line 48-50.
Between the exclusion criteria, you should add to not have at least one of the criteria of MEts according to the NCEP.
Please add the number of ethical approval within the text.
I am not sure that lines 202-207 are part of the study.
Line 222 is not clear: the other nurse were not graduated?
There is a shift of the numbers of lines from 251 to 256 in the left side of the paper and into table 4. Please fix it.
Line 308, due to the small sample size, it is better to say that "the study suggests...." because it is not possible to generalize to much the present results (such as you wrote in lines 313-315).
I don't understand the utility of lines 310-312 in this specific part.
Please add the limitations of the study.
In lines 331 you wrote that "The research was funded by Chang Gung Medical Fundation" and in line 346 you wrote "The author declare no funding". Please clarify this.
Reviewer 2 Report
This study found an interesting questions based nurse sample. I appreciate the work the authors have done. However, many questions existed in this study blocked the potential publication.
1. The abstract part is not concise. Why you used the the transtheoretical model? What is the benefits?Are there any theoretical contributions? You should not only talk about your discovery, but also about what the meaning of your discovery is.
2. The second and third paragraphs, you just listed the current literature. Where are the summaries and gaps in existing research? There is a lack of corresponding summaries.
3. TTM model in introduction is too simply. Why you use this model instead of others? Please specify.
4. I noticed that you mentioned Taiwan many times in introduction part and discussion part. Taiwan nurses are just your research background. Are there any differences between Taiwanese nurses and nurses from other regions?
5. Is your experiment ethically justified?
6. Table 1/2/3, the strategies part is too long. This makes the table is not “artistic”. I strongly recommend that the author move these words into the text and keep only the corresponding points here.
7. Can you draw a graph to show the differences in change? In this version PDF, I cannot capture your key points in the tables.
8. I suggest the authors to compare your studies to other similar studies to highlight your contributions.
Round 2
Reviewer 1 Report
I thank the authors for editing and implementing the article, now, in my opinion, is ready for publication.
Reviewer 2 Report
Thanks for your revision. I think this version can be accepted